# Fabrication of the Zirconium Diboride-Reinforced Composites by a Combination of Planetary Ball Milling, Turbula Mixing and Spark Plasma Sintering

**DOI:** 10.3390/ma14144056

**Published:** 2021-07-20

**Authors:** Iwona Sulima, Paweł Hyjek, Marcin Podsiadło

**Affiliations:** 1Institute of Technology, Pedagogical University of Krakow, Podchorazych 2 St., 30-084 Krakow, Poland; pawel.hyjek@up.krakow.pl; 2Łukasiewicz Research Network—Krakow Institute of Technology, Zakopiańska 73, 30-418 Kraków, Poland; marcin.podsiadlo@kit.lukasiewicz.gov.pl

**Keywords:** planetary ball milling, turbula mixing, composites, zirconium diboride, spark plasma sintering

## Abstract

The aim of this study was to carry out the consolidation of zirconium diboride-reinforced composites using the SPS technique. The effect of the adopted method of powder mixture preparation (mixing in Turbula or milling in a planetary mill) and of the reinforcing phase content and sintering temperature on the microstructure, physical properties, strength and tribological properties of sintered composites was investigated. Experimental data showed that the maximum relative density of 94–98% was obtained for the composites sintered at 1100 °C. Milling in a planetary mill was found to contribute to the homogeneous dispersion and reduced clustering of ZrB_2_ particles in the steel matrix, improving in this way the properties of sintered steel + ZrB_2_ composites. Morphological and microstructural changes caused by the milling process in a planetary mill increase the value of Young’s modulus and improve the hardness, strength and wear resistance of steel + ZrB_2_ composites. Higher content of ZrB_2_ in the steel matrix is also responsible for the improvement in Young’s modulus, hardness and abrasive wear resistance.

## 1. Introduction

The development of modern technology and the needs of industry require constant development of new materials with unique properties, the production of which, by conventional methods, is either impossible or very difficult. Such materials also include composites, whose properties depend, among others, on the method of fabrication and appropriate selection of materials used for the composite matrix and reinforcing phase. The composite material is a combination of at least two different materials which operating jointly provide better properties than when used individually [1,2]. Recent decades have faced an intensive development of research on metal matrix composites. The most commonly used materials for the composite matrix are aluminum, iron, titanium, copper, magnesium, and their alloys [3,4,5,6,7]. Typical reinforcements of composites are carbides (TiC, SiC, B_4_C), oxides (ZrO_2_, TiO_2_, Al_2_O_3_, SiO_2_), nitrides (Si_3_N_4_, BN) and borides (TiB_2_, ZrB_2_) [8,9,10,11]. Very vivid interest in the use of zirconium diboride (ZrB_2_) as a reinforcing phase for metal matrix composites results from a unique combination of physical and mechanical properties offered by this ceramic material [12,13,14,15]. The beneficial properties of zirconium diboride include high melting point (3245 °C), high hardness (>2200 HV), excellent thermal conductivity (57.9 W/mK) and electrical conductivity (1 × 10^7^ S/m). Additionally, ZrB_2_ has good corrosion resistance [16,17,18]. Various experiments have shown that the use of ZrB_2_ ceramic particles can significantly improve the mechanical properties and wear behaviour of aluminum and aluminum matrix composites [19,20,21]. The beneficial effect of ZrB_2_ on hardness, strength and electrical conductivity was also observed in composites based on copper matrix [12,22]. For example, Kumar et al. [21] showed an improvement in the mechanical properties of aluminum matrix composites along with an increase in the content of zirconium diboride. In turn, Wang et al. [12] produced Cu–ZrB_2_ composites by the method of hot sintering with hardness higher than 100 HV and electrical conductivity higher than 85% IACS.

Using iron and its alloys for the composite matrix is of great importance for various industries. Iron alloys are characterized by good mechanical properties, high wear resistance and low production costs. The use of stainless steel is preferable to iron due to its higher corrosion resistance and better mechanical properties [23,24,25]. Studies devoted to the development of sintered composites with an austenitic steel matrix have mainly concerned the selection of sintering process parameters and characteristics of the physical, mechanical and functional properties of such materials [26,27,28,29,30,31]. Nahme et al. [30] investigated the microstructure and mechanical properties of sintered steel-based composites reinforced with 15 vol% TiB_2_. It has been indicated that the use of fine ceramic particles (TiB_2_) improves the mechanical properties of composites. Such materials usually show a notable increase in both compressive strength and tensile strength compared to the alloy without reinforcement. Studies of the composite microstructure showed homogeneous distribution of TiB_2_ in the steel matrix. Sulima et al. [15] investigated the effect of the content of ZrB_2_ on the density, microstructure and properties of composites based on 316 L austenitic stainless steel matrix. With the increasing content of ZrB_2_, an increase in Young’s modulus and hardness was obtained. The incorporation of ZrB_2_ into the matrix also had a beneficial effect on the wear resistance of the composites. Shamsuddin et al. [31] sintered Fe–Cr–Al_2_O_3_ composites with different proportional content of the reinforcing phase (5%–25 wt%) using a free sintering process. The addition of 5 wt% Al_2_O_3_ increased the relative density as compared to the samples without the reinforcing phase. On the other hand, the increase in Al_2_O_3_ content from 10 wt% to 25 wt% reduced the density. The results showed that both hardness and wear resistance of the sintered composites improved when increasing the content of the reinforcing phase by up to 20 wt%. Further increase in the Al_2_O_3_ content reduced the mechanical properties due to the alloy’s tendency to form agglomerates. Akhtar et al. [32] used MoSi_2_ as a reinforcing phase for 316 L austenitic steel. The addition of MoSi_2_ to the steel matrix increased the relative density, hardness and tensile strength of the composites. The best combination of the properties was obtained for the composite with the addition of 5 wt% MoSi_2_ sintered at 1300 °C. In [33], the properties of sintered composites based on a steel matrix (AstaloyCrL) reinforced with tungsten carbide (5 vol% and 20 vol%) were examined. It was shown that the volume fraction of carbides controls both abrasive wear resistance and corrosion resistance of composites. With the growth of the WC phase, the composites become more susceptible to pitting corrosion. Higher content of tungsten carbide in the composite increases the coefficient of friction and reduces the weight loss of the sample. Moreover, it has been found that the size of the particles of the reinforcing phase has no significant effect on the coefficient of friction and corrosion resistance.

The main purpose of this study was to determine the effect of the powder mixture preparation method on the properties and microstructure of composites based on the 316 L austenitic stainless steel matrix reinforced with different contents of the ZrB_2_ phase.

## 2. Test Materials and Methods

The starting materials were 316 L austenitic stainless steel (Hoganas, 25 μm average particle size) and zirconium diboride (H.C. Starck, 99.9 wt% purity, 2.5–3.5 μm average particle size). Table 1 shows the chemical composition of the 316 L steel powder, while Figure 1 shows the morphology of the 316 L steel and ZrB_2_ powders.

For the sintering process, two mixtures of composite powders with different content of the ZrB_2_ reinforcing phase were prepared:316 L steel + 5 wt% ZrB_2_316 L steel + 10 wt% ZrB_2_

Two methods were used for the preparation of powder mixture:Dry mixing in a Turbula T-2C mixer (Willy A. Bachofen AG, Muttenz, Switzerland) for 8 h. Steel balls with a diameter of 5 mm were used to homogenize each mixture.Milling in a Fritsch Pulverisette 6 planetary mill (Fritsch GmbH -Germany) using a 5 mm diameter milling beaker and balls made of tungsten carbide. The rotational speed of the grinder was 200 rpm and the milling time was 8 h. A solution of polyethylene glycol in ethanol was used as the wetting medium. The milling balls:ground powder weight ratio was 10:1.

Figure 2 shows the morphology of composite mixtures containing 5 wt% and 10 wt% ZrB_2_ prepared by two different methods.

The powders were consolidated by spark plasma sintering (SPS HP5 FCT System, Frankenblick, Germany) in a graphite die (Ø 20 mm) at temperatures of 1000 °C and 1100 °C. The temperature was controlled by a pyrometer. Sintering experiments were conducted in an argon atmosphere at 35 MPa pressure for 5 min. During SPS process, the heating rate was kept at a level of 200 °C/min. The cooling rate of the furnace was 100 °C/min. The sintered samples were of 20 mm diameter and 6 mm height.

The density and open porosity of the sintered composites were determined by the Archimedes’ immersion method in water [34]. In the study of the Young’s modulus of the sintered composites, a Panametrics Epoch III flaw detector (Billerica, MA, USA) equipped with special broadband ultrasound probes for longitudinal and transverse waves was used. Five measurements were made for each sample. The measurement error was 2%.

The microstructures were analyzed by Scanning Electron Microscopy (SEM, JEOL JSM 6610LV (JOEL, Peabody, MA, USA) and Hitachi SU-70 (Hitachi High-Technologies Corporation, Tokio, Japan), Energy Dispersive Spectroscopy (EDS, AZtec) and Wavelength Dispersive Spectroscopy (WDS). The phase identification was performed by X-ray diffraction method (XRD, Empyrean; PANalytical) using CuKα radiation. Additionally, phase analysis was coupled with SEM using electron backscatter diffraction (EBSD) analysis. The applied SEM microscope was a FEI Inspect F50 (FEI Company, Hillsboro, OR, USA) with an EDAX TSL EBSD detector. The obtained EBSD data was analyzed using TSL OIM software [35] using a grain tolerance angle of 5 degrees and a PDF ICCD 2011 database format.

Vickers hardness (HV1) was evaluated under a load of 9.81 N using a NEXUS 4000 tester (INNOVATEST EUROPE BV, Maastricht, The Netherlands). The hardness values were the average of ten readings taken at random locations in the sintered sample. Next, the mechanical properties were determined in tensile tests and compression tests. The tensile tests were carried out at room temperature on specimens with a base of 5 mm and cross-section base of 0.5 × 0.5 mm. The samples were deformed at a rate of 6 × 10^−4^ s^−1^. Three specimens were tested for each material. The compression tests were conducted using an INSTRON TT-DM testing machine (Norwood, MA, USA). Cylindrical specimens with a diameter of 3 mm and a height of 4.5 mm were used. Compression tests were carried out with the crosshead speed of 1 × 10^−4^ mm/s at room temperature and at temperatures of 400 °C and 800 °C. Studies at elevated temperatures were performed in a protective argon atmosphere to prevent oxidation of compression anvils and specimens. Three specimens were tested for each material.

Tribological tests were carried out using a ball-on-disc wear testing machine (ELBIT, Koszyce Małe, Poland). Tests were carried out without lubricant according to the ISO 20808:2004(E) [36]. Table 2 shows the conditions used during tribological tests. For each test a new ball was applied. All balls and samples were washed in high-purity acetone and dried.

During the test, the friction force was continuously measured with an extensometer. The friction coefficient (μ) was calculated as the ratio of the friction force (*F_n_*) and the applied load (*F_t_*). Next, the specific wear rate was calculated. The cross-sectional profile of the wear track was measured at four places at intervals of 90° using a contact stylus profilometer. The specific wear rate was calculated from the following equation:(1)$$W_{V\left(disc\right)}=\frac{V_{disc}}{F_{n}\cdot L}$$
where:

*V_disc_*—wear volume of disc specimen [mm^3^],

*F_n_*—applied load [N],

*L*—sliding distance [m].

## 3. Results and Discussion

It is the fact well known that the method of powder mixture homogenization significantly affects the quality of the sintered composite. In the preparation of composite powder mixtures, the choice of the mixing method is one of the main issues [37,38,39].

The differences in the morphology of composite powders after milling in a planetary mill or mixing in Turbula are shown in Figure 2. Milling processes change the morphology of powders as a result of the strong plastic deformation to which the particles are subjected during milling. The particles of the composite mixture have an elongated shape (Figure 2c,d).

Figure 3 shows the microstructure of the sintered 316 L steel and steel + ZrB_2_ composites. Microscopic observations have revealed qualitative differences in the morphology of the examined microstructures. A heterogeneous distribution of the ZrB_2_ reinforcing phase in the steel matrix (Figure 3b,c) was observed in the case of mixing in Turbula. Zirconium diboride tended to be unevenly located along the grain boundaries in the matrix (Figure 3b,c). The formation of agglomerates of the reinforcing phase was observed locally (Figure 3c). For these composites, the size of the ZrB_2_ reinforcing phase was in the range of 3–12 µm. In contrast, during mechanical milling in a ball mill, the size of the ZrB_2_ phase was reduced. The size of the reinforcing phase was comprised in the range of 1–3 µm. The reinforcing phase was evenly distributed in the matrix, especially in the case of composites containing 10%ZrB_2_ (Figure 3e). The formation of agglomerates of the reinforcing phase in the microstructure was not observed. In the steel-5%ZrB_2_ composites (Figure 3d), local occurrence of areas free from the presence of the reinforcing phase was noticed. The fragmentation of the ceramic phase was due to the operation of mechanisms inducing changes in the morphology of particles during mechanical milling process. These mechanisms included: (a) mechanism of plastic deformation, (b) mechanism of cold welding, and (c) fracture mechanism. In the first stage of the milling process, the powder particles were sliding on top of each other due to the effect of cracking and plastic deformation. Powders underwent plastic deformation which resulted in their hardening and then cracking. As a result of cracking, new surfaces were created, which in the second stage of the process allowed for cold welding of the powder particles. In the third stage, the powder particles were strongly deformed and crushed [38,40].

Figure 4 shows the X-ray diffraction patterns of materials sintered at 1100 °C. The phase composition analysis revealed the presence of the ZrB_2_ phase and 316 L austenitic stainless steel. It was observed that the intensity of the peaks of the reinforcing phase increased with the increased content of the ZrB_2_ particles in the steel matrix. A similar tendency was observed for composites sintered at 1000 °C.

Figure 5, Figure 6, Figure 7, Figure 8, Figure 9 and Figure 10 show the comparative microstructures of the sintered composites. WDS and EBSD techniques were used to verify the phase composition of the sintered steel + ZrB_2_ composites. The results of phase analysis confirmed the presence of the ZrB_2_ phase in all sintered composites. Microstructural studies in turn confirmed the differences in the size and morphology of the reinforcing phase depending on the method of preparation of powder blends. A significant fragmentation of the reinforcing phase was observed for composites which were processed in a planetary mill. The fine reinforcing phase was evenly distributed in the entire volume of the steel matrix (Figure 7 and Figure 8). This was characterized by an oval or spherical shape. Additionally, for all steel + ZrB_2_ composites, the EBSD analysis (Figure 9 and Figure 10) showed the presence of very fine precipitates containing chromium, which were not observed in the sintered austenitic steel. The WDS chemical composition analysis has indicated that the precipitates contained both chromium and boron (Figure 5, Figure 6, Figure 7 and Figure 8). Probably, the applied conditions of the SPS process promoted the formation of new borides in the microstructure of composites. During the SPS process, several mechanisms operate simultaneously (surface activation, diffusion, fusion, necking between the sintered powder particles, and plastic flow), and they may contribute to the formation of new phases in the microstructure of steel + ZrB_2_ composites. In composites processed by mixing in Turbula (Figure 5 and Figure 6), the chromium- and boron-containing phases were distributed along the grain boundaries in the matrix.

When milling was carried out in a planetary mill, the examined chromium and boron dispersion phases were evenly distributed in the grains of the steel matrix and along the grain boundaries, acting as an additional reinforcement of the steel matrix (Figure 8). Additionally, the formation of nickel- or molybdenum-containing precipitates was locally observed in the microstructure of all steel + ZrB_2_ composites.

This is consistent with the results of tests carried out on the TiB_2_-reinforced composites sintered by SPS [8,41]. In the microstructure of these materials, the presence of two complex borides of the BCr_0.2_Fe_1.8_ and (Cr,Fe,Mo,Ni,Ti)_3_B_2_ type was found. Studies of the microstructure [41] also showed that changes in the conditions of the SPS process have a significant impact on the microstructure of the tested materials. Longer time and higher temperature of the sintering process increase the number of new boride phases and their dimensions in the entire volume of the composite. Attention deserves the fact that, as indicated in [42,43], the addition of boron effectively activates the free sintering process of austenitic steel. The formation of complex borides of the (Cr,Mo,Fe)_2_B type during free sintering of austenitic steel was demonstrated. It was observed that during sintering of the boron-modified austenitic steel, a liquid phase was formed as a result of the eutectic reaction between matrix alloy and complex borides.

Figure 11 shows the effect of powder preparation method and reinforcing phase content on the density and Young’s modulus of the sintered composites. The results obtained indicate that milling in a planetary mill is the method more preferable, since all composites have higher density and Young’s modulus compared to the results obtained for the Turbula mixing method. The results also show that, regardless of the method of powder preparation, the sintering temperature of 1000 °C is not sufficient for the consolidation of composites with an austenitic steel matrix. The sinters obtained under such conditions are characterized by low density, i.e., at a level of 79–89%, and high porosity in the range of 8–20%. Low Young’s modulus values (Figure 11b) were also obtained for these materials because of the high porosity of sinters fabricated at this temperature.

Significant improvement in the density and Young’s modulus of the sintered materials was obtained when the sintering temperature was raised to a higher level. All composites sintered at 1100 °C were characterized by a very high degree of compaction. The density was in the range of 94–98% of the theoretical density. For these materials, the minimum porosity was in the range of 0.4–1.3%. The higher the sintering temperature, the easier and more intense the diffusion process, improving the composite density. The increase in sintering temperature also raised the value of Young’s modulus (Figure 11b). Additionally, it was observed that the density of the sintered composites decreased when increasing the content of ZrB_2_. This was due to the lower density of zirconium diboride (6.08 g/cm^3^) [44] compared to the density of 316 L steel (8.00 g/cm^3^) [45].

Figure 12 shows the results of hardness tests. Significant improvement in hardness was observed in all composites sintered at 1100 °C, additionally enhanced by milling the composite mixtures in a planetary mill. For comparison, the hardness of 316 L steel without reinforcement is 197 HV1 but for composites with 5% and 10% ZrB_2_, an almost two-fold increase in hardness was obtained, i.e., to 353 HV1 and 395 HV1, respectively. Salur at al. [46] showed that, compared to the starting alloy (composite matrix), the hardness of the TiC-reinforced composites increased three times when milling in a ball mill was applied. A homogeneous distribution of nanoparticles in the matrix was observed with the increasing time of milling. The analysis of the effect of the content of the reinforcing phase showed an obvious improvement in the hardness of composites. The hardness was increasing with the increasing content of the ZrB_2_ phase. This tendency was observed in composites sintered under various conditions, and it is consistent with the research done by other authors [47,48], who have demonstrated that hardness of the 304 austenitic steel matrix increases with the increasing volume fraction of ZrB_2_.

The results obtained so far for properties such as density, porosity, Young’s modulus and hardness clearly indicate that the sintering temperature of 1000 °C is insufficient. Therefore, in the further part of discussion, the results obtained for the mechanical and tribological properties will concern only materials sintered at a temperature of 1100 °C.

Figure 13 and Table 3 compare the results of compression tests carried out on the sintered materials at room temperature and also at elevated temperatures (400 °C and 800 °C). Figure 14 shows the compression samples before and after the test. The test results indicate that, after reaching the maximum true stress, the stress value gradually decreases for each tested material. Cracks and fractures were not observed in the samples that underwent plastic deformation during the test. The samples were deformed to the appropriate level of strain. Fractures appeared in samples compressed at 400 °C and 800 °C (Figure 14). The results showed that changes in the content of the reinforcing phase in the matrix had no important effect on the mechanical properties of the composites in the entire range of compression test temperatures. Only in the compression tests carried out at room temperature (Figure 13a) was the maximus true stress observed to assume higher values for the composites containing 10% ZrB_2_. A similar effect was observed in other studies, where the steel matrix was reinforced with oxides (Y_2_O_3_) and carbides (TiC) [49,50].

The results obtained (Figure 13, Table 3) show that, compared to steel without reinforcement, various methods used for the composite mixture preparation have a significant effect on the improvement in mechanical properties of the tested composites. This effect is clearly visible in the results obtained at room temperature and at 400 °C (Figure 13a,b). The results of the compression tests carried out at room temperature have demonstrated that, for milling in a planetary mill, the compressive strength measured as the maximum force F divided by the initial cross-sectional area is 1139 MPa and 1189 MPa for the steel-matrix composites containing 5%ZrB_2_ and 10%ZrB_2_, respectively. For comparison, the compressive strength of the austenitic steel is 753 MPa.

The effect of composite hardening can be obtained by introducing the ceramic phase into the matrix, and by either inhibiting the grain growth or refining the matrix grains [51]. Owing to the simultaneous application of the ZrB_2_ reinforcing phase and its fragmentation during milling, an effective reinforcement of the composites was obtained with further improvement in their strength properties determined during compression tests. Moreover, the compressive stresses formed in the material during the first stage of compression contribute to closing of pores in the sintered material and may delay the initiation of cracks at the matrix/reinforcement interface [52].

Compression tests at elevated temperatures (400 °C and 800 °C) deteriorated the strength of the sintered materials (Figure 13 b,c, Table 3). As a result of these tests, regardless of the temperature at which they were carried out, a nearly identical course of the deformation curves was obtained for each group of the tested materials. In the analysis of the results of compression tests, the content of the reinforcing phase was found to have no major effect on the compressive strength. The steel + ZrB_2_ composites tested at 400 °C were characterized by a compressive strength in the range of about 810–820 MPa and 594–603 MPa for milling in a planetary mill and mixing in Turbula, respectively. For comparison, the compressive strength of the 316 L austenitic steel was 529 MPa. In contrast, the application of higher temperature in compression tests, i.e., 800 °C, drastically reduced the strength of all materials tested (Figure 13c, Table 3).

Studies of the mechanical properties conducted by tensile tests also showed that the use of the milling method allows for the fabrication of composite materials with high plasticity and good strength properties at room temperature (Figure 15). The tensile strength R_m_ reached the values of 832 MPa and 730 MPa for the steel + 5%ZrB_2_ composite and steel + 10%ZrB_2_ composite, respectively. This demonstrates an increase in the tensile strength compared with the sintered 316 L steel for which the tensile strength R_m_ of 690 MPa was obtained. At the same time, along with the increasing content of ZrB_2_ in the steel matrix, the plasticity was decreasing. For comparison, the elongation of 316 L steel was 32%, while for the composites with 5%ZrB_2_ and 10%ZrB_2_ it decreased to 22% and 19%, respectively.

This is consistent with the research carried out by Tjong and Lau [48], who have demonstrated that the addition of titanium diboride improves the mechanical strength of the 304 austenitic steel-based composite at the expense of plasticity. Other research works [30] have also confirmed that introducing the TiB_2_ ceramic particles (15 vol%) with a size of less than 10 µm into the 316 L steel matrix is a good way to improve the mechanical properties. The tensile strength of the sintered steel was 520 MPa, while the tensile strength of the composites increased to 885 MPa. Studies also showed that the deformation decreased from 45% (for 316 L steel) to 6% (for steel + 15TiB_2_ composite).

Tests of the abrasive wear resistance were carried out at room temperature by the ball-on-disc method. Changes in the coefficient of friction and wear index as a function of the ZrB_2_ content are shown in Table 4. The test results obtained show that the addition of the ZrB_2_ ceramics to the steel matrix improves the tribological properties of composites. The coefficient of friction and the wear index decrease along with increasing volume fraction of ZrB_2_ in the steel matrix. The lowest values of the coefficient of friction (0.47 and 0.4) and specific wear rate (16.9 × 10^−6^ mm^3^/Nm and 13.1 × 10^−6^ mm^3^/Nm) were obtained for the composites with 10% ZrB_2_, taking into account various methods of the powder mixture preparation. For comparison, the coefficient of friction and wear rate of the 316 L steel were 0.64 and 32.5 × 10^−6^ mm^3^/Nm, respectively. The reinforcing phase protects the steel matrix from the effect of friction and reduces the wear rate. As the content of ZrB_2_ increases, the loss of composite material is reduced. These results are consistent with the literature [48], where it has been demonstrated that adding TiB_2_ to the matrix significantly improves the abrasive wear resistance of composites based on 304 steel. Srivastava et al. [53] showed that composites reinforced with TiC and (Ti,W)C exhibited better wear resistance than the austenitic steel without reinforcement. Composites containing (Ti,W)C were characterized by abrasive wear resistance superior to that obtained in the composites reinforced with TiC. The effect of the TiAl reinforcing phase content (3%–9 vol%) on the coefficient of friction and abrasive wear of sintered composites based on the 316 L austenitic steel matrix was investigated in [54]. It has been found that the wear rate of composites decreases with the increasing content of the TiAl intermetallic phase and sintering temperature.

The conclusion is that the tribological properties depend on the method of preparation of composite powders (Table 4). Milling in a planetary mill reduces the coefficient of friction, and thus the wear rate of the material, compared to the results obtained for composites mixed in Turbula. This is due to the microstructure of composites, which is characterized by a homogeneous distribution of the fine ZrB_2_ reinforcing phase in the entire volume of the matrix. The additional effect is the formation of very fine phases with chromium and boron, which are evenly distributed in the grains of the steel matrix and along the grain boundaries. They provide an additional reinforcement to the steel matrix (Figure 8), and this has a positive effect on the abrasive wear resistance of the tested composites.

As a result of tribological tests, wear tracks were formed. Sample wear tracks are shown in Figure 16. In all sintered composites, the area of wear tracks showed signs characteristic of the abrasive and adhesive wear, such as scratches and grooves of orientation consistent with the direction of ball movement (bright arrow on the image of microstructure) and delamination. During tests, in the area of wear, permanent deformation of the composite material combined with abrasion occurred.

## 4. Conclusions

The aim of this study was to investigate the effect of milling in a planetary mill and mixing in Turbula on the microstructure and resultant physical and mechanical properties of the steel + ZrB_2_ composites. The important conclusions are as follows:The test results showed that the sintering temperature of 1000 °C and the SPS method are not sufficient to produce a composite material with high properties.The microstructure of the sintered composites depends on the method of powder preparation. A heterogeneous distribution of the ZrB_2_ reinforcing phase in the steel matrix was observed when Turbula mixing was used. Milling in a planetary mill contributed to the refinement and homogeneous dispersion (distribution) of the ZrB_2_ particles in the matrix. Microstructural examinations additionally revealed the presence of numerous fine precipitates containing chromium and boron.The increase in the weight fraction of ZrB_2_ increased Young’s modulus, hardness and abrasion resistance.The method of preparing powders significantly affects the properties of the steel + ZrB_2_ composites. Milling in a planetary mill is beneficial because all tested composites showed an improvement in density and an increase in Young’s modulus, hardness, strength and wear resistance.The most advantageous combination of physical, mechanical and tribological properties was obtained for the steel + 10%ZrB_2_ composites processed by milling in a planetary mill and sintered by SPS at a temperature of 1100 °C for the time of 5 min.

## Figures and Tables

**Figure 1 materials-14-04056-f001:**
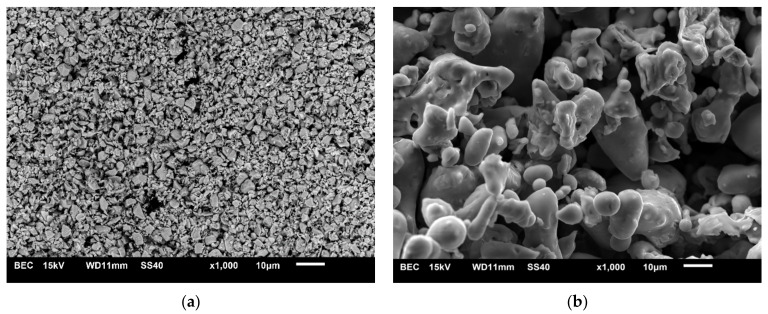
SEM images of the starting powders of (**a**) ZrB_2_ and (**b**) 316 L austenitic stainless steel.

**Figure 2 materials-14-04056-f002:**
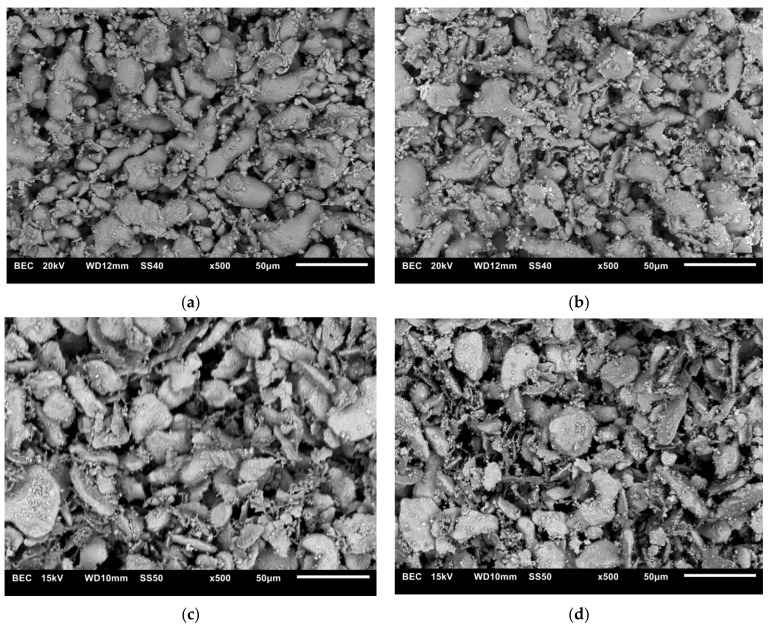
SEM images of powder mixtures of: (**a**) 316 L steel + 5 wt% ZrB_2_ (Turbula), (**b**) 316 L steel + 10 wt% ZrB_2_ (Turbula)_,_ (**c**) 316 L steel + 5 wt% ZrB_2_ (planetary mill), (**d**) 316 L steel + 10 wt% ZrB_2_ (planetary mill).

**Figure 3 materials-14-04056-f003:**
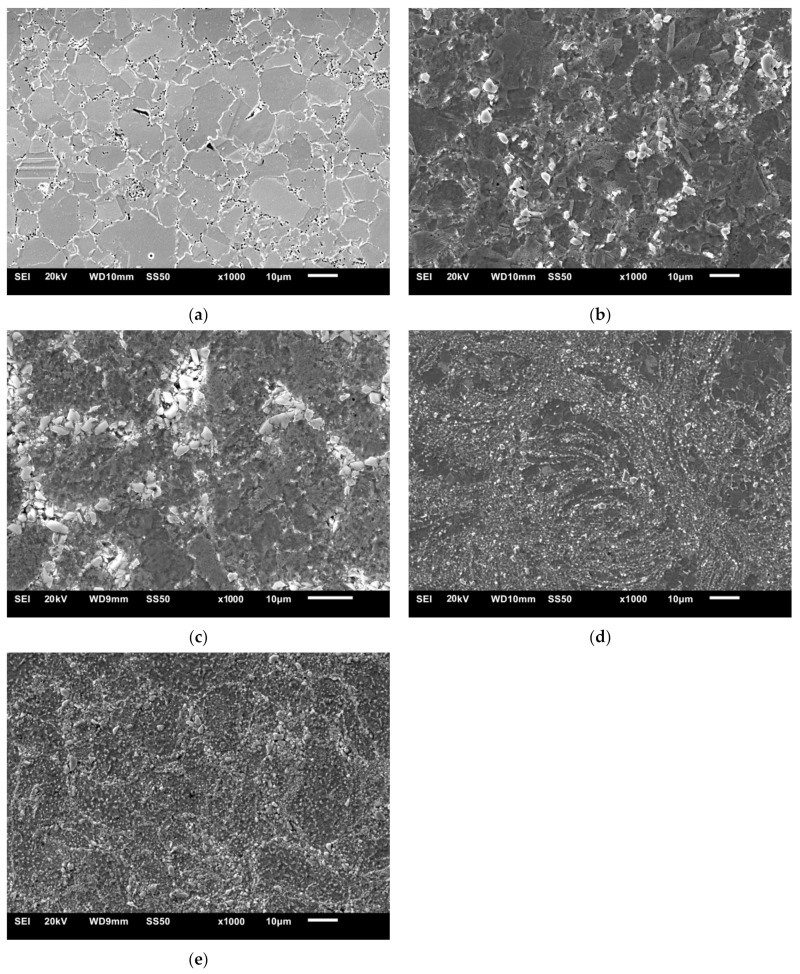
The SEM micrograph of: (**a**) 316 L austenitic stainless steel, (**b**) steel + 5%ZrB_2_ composite (Turbula), (**c**) steel + 10%ZrB_2_ composite (Turbula), (**d**) steel + 5%ZrB_2_ composite (planetary mill), and (**e**) steel + 10%ZrB_2_ composite (planetary mill).

**Figure 4 materials-14-04056-f004:**
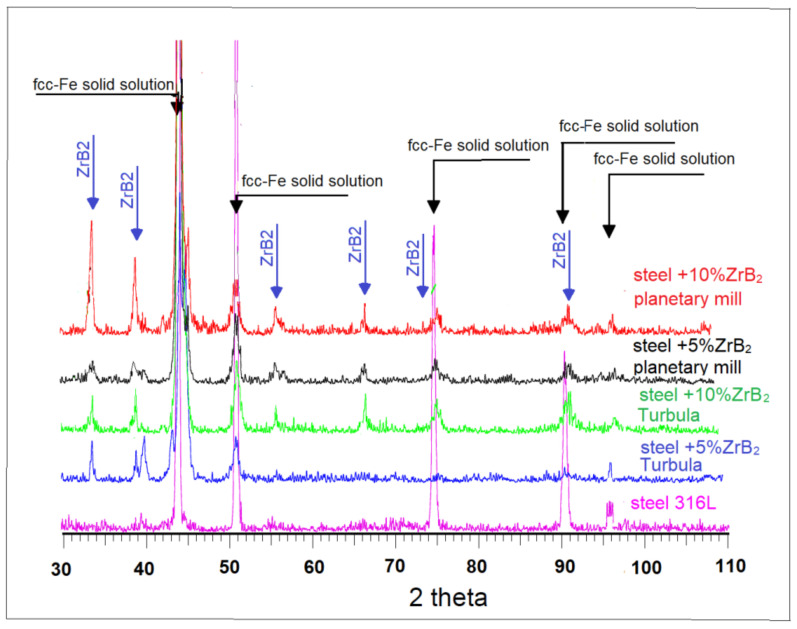
The XRD diffraction patterns of 316 L steel +ZrB_2_ composites sintered at 1100 °C.

**Figure 5 materials-14-04056-f005:**
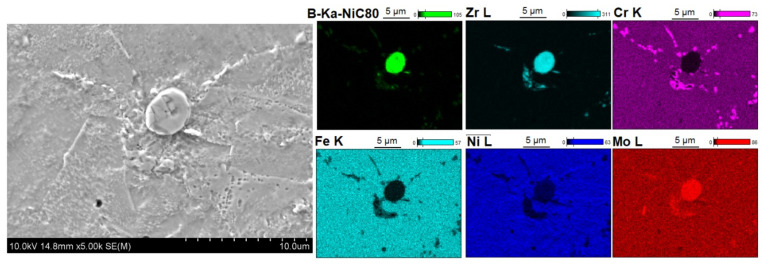
The microstructure (SEM) of steel + 5%ZrB_2_ composite (Turbula, SPS 1100 °C) with corresponding area analysis (WDS).

**Figure 6 materials-14-04056-f006:**
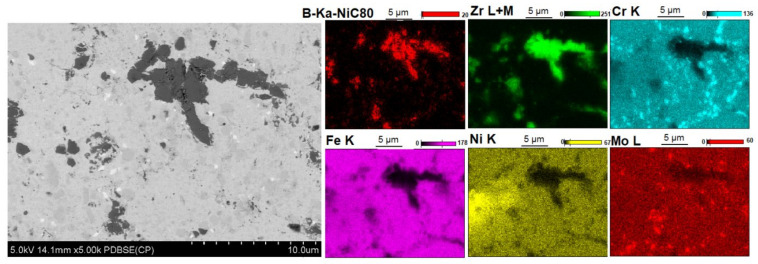
The microstructure (SEM) of steel + 10%ZrB_2_ composite (Turbula, SPS 1100 °C) with corresponding area analysis (WDS).

**Figure 7 materials-14-04056-f007:**
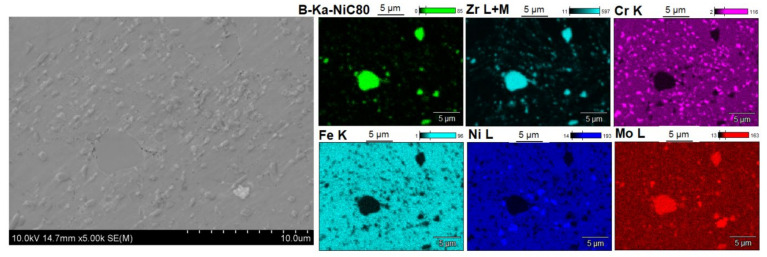
The microstructure (SEM) of steel + 5%ZrB_2_ composite (planetary mill, SPS 1100 °C) with corresponding area analysis (WDS).

**Figure 8 materials-14-04056-f008:**
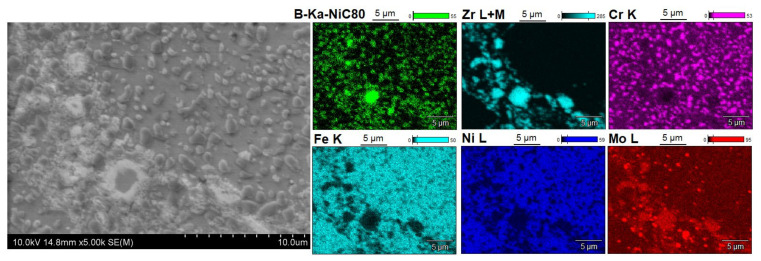
The microstructure (SEM) of steel + 10%ZrB_2_ composite (planetary millSPS 1100 °C) with corresponding area analysis (WDS).

**Figure 9 materials-14-04056-f009:**
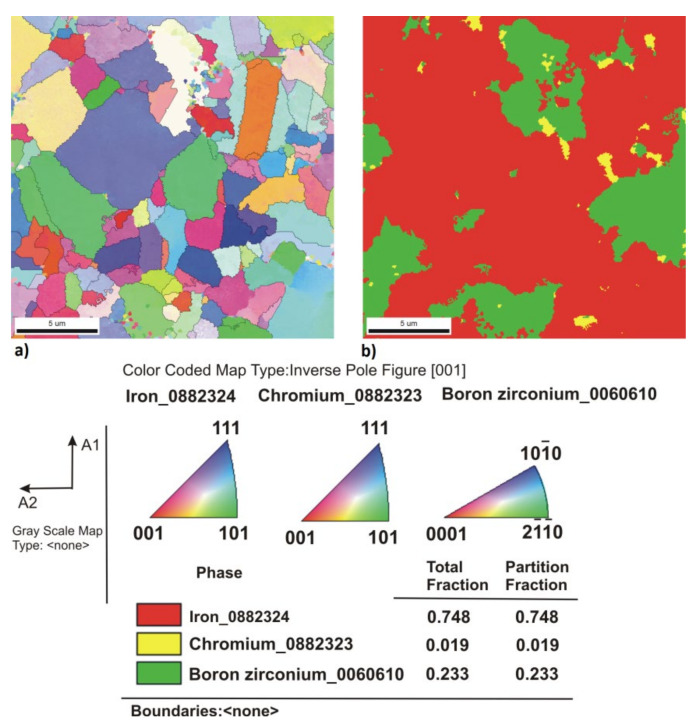
The results of EBSD analysis of the steel + 10%ZrB_2_ composite sintered at 1100 °C (Turbula): (**a**) the crystallographic orientation of grains, and (**b**) phase analysis (Chi-Scan).

**Figure 10 materials-14-04056-f010:**
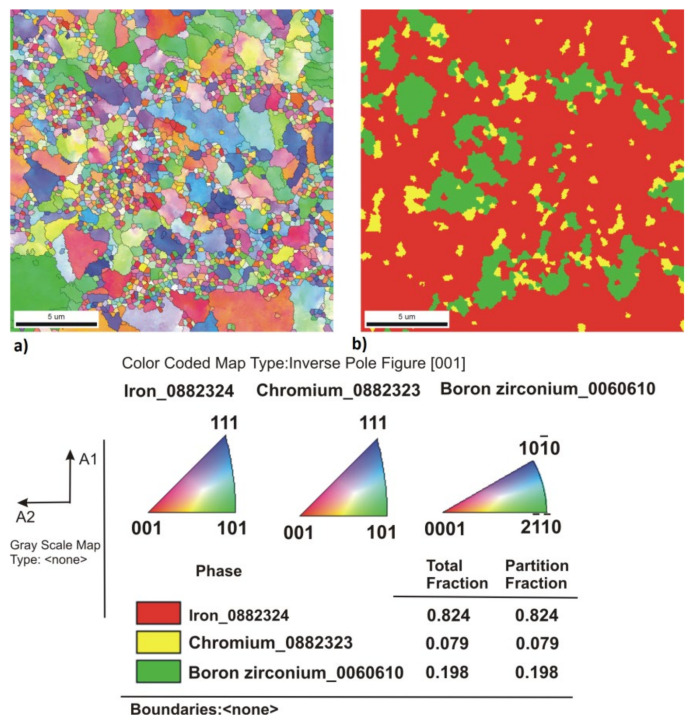
The results of EBSD analysis of the steel + 10%ZrB_2_ composite sintered at 1100 °C (planetary mill): (**a**) the crystallographic orientation of grains, and (**b**) phase analysis (Chi-Scan).

**Figure 11 materials-14-04056-f011:**
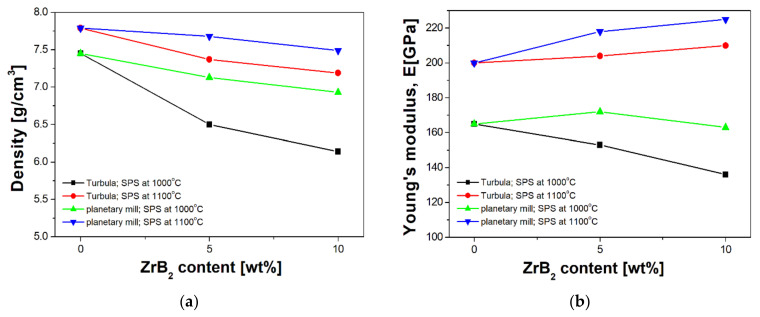
The density (**a**) and Young’s modulus (**b**) of consolidated materials as a function of ZrB_2_ content.

**Figure 12 materials-14-04056-f012:**
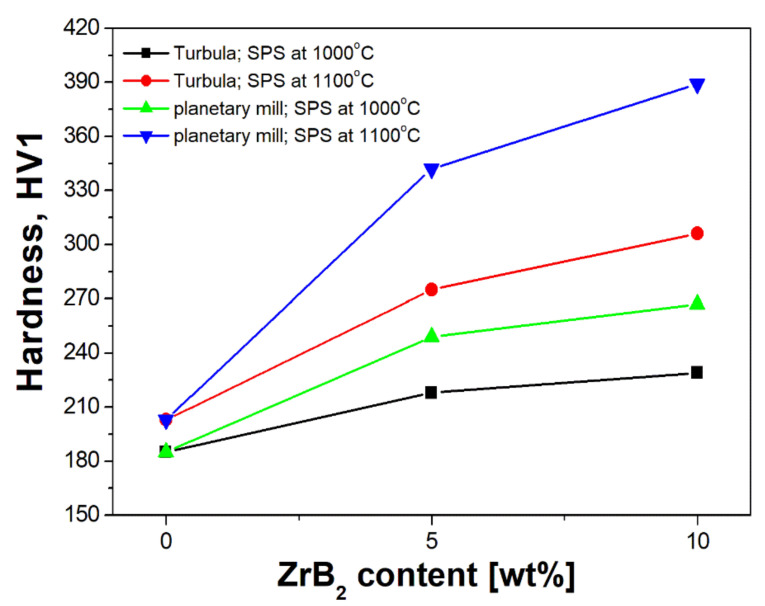
Hardness *HV1* of consolidated materials as a function of ZrB_2_ content.

**Figure 13 materials-14-04056-f013:**
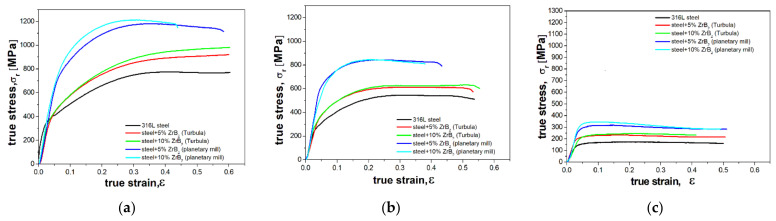
The results of compression tests carried out at (**a**) room temperature and at temperatures of: (**b**) 400 °C and (**c**) 800 °C for spark plasma sintered materials.

**Figure 14 materials-14-04056-f014:**
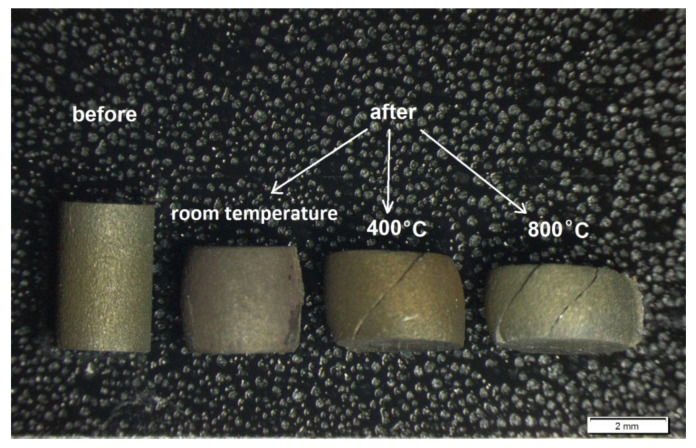
Image of sintered samples before and after the compression tests.

**Figure 15 materials-14-04056-f015:**
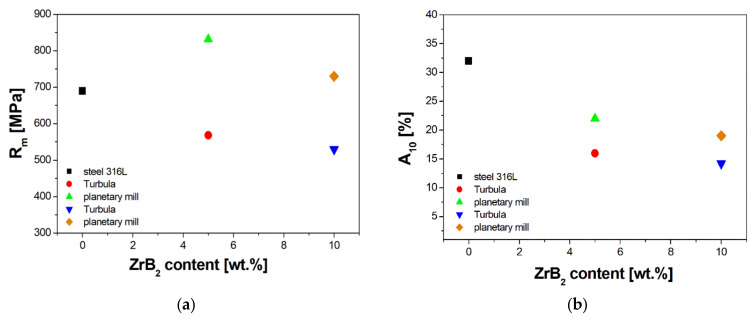
The mechanical properties: (**a**) tensile strength (R_m_) and (**b**) elongation (A_10_) determined during the tensile test of 316 L steel and steel + ZrB_2_ composites.

**Figure 16 materials-14-04056-f016:**
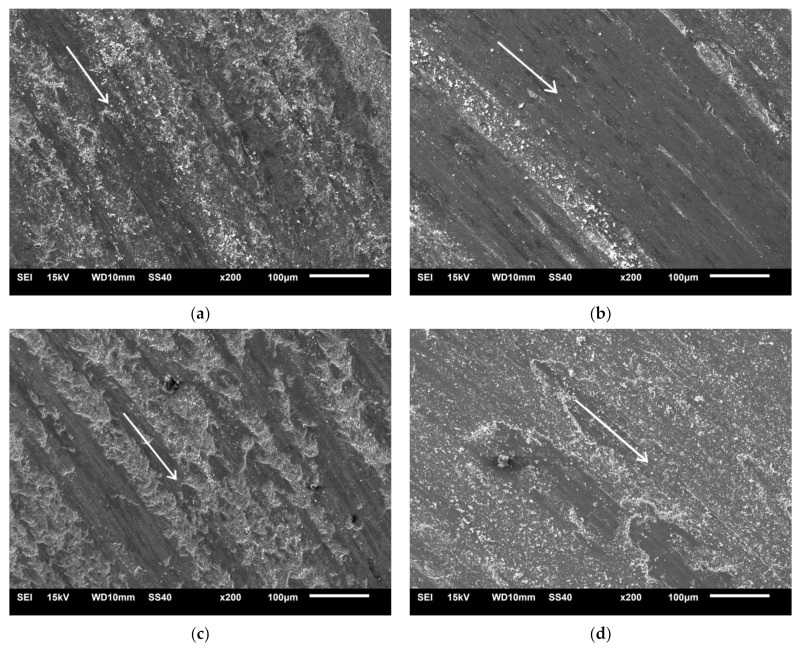
Surfaces worn by the wear test: (**a**) steel + 5% ZrB_2_ composite (Turbula), (**b**) steel + 10% ZrB_2_ composite (Turbula) (**c**) steel + 5% ZrB_2_ composite (planetary mill), and (**d**) steel + 10% ZrB_2_ composite (planetary mill).

**Table 1 materials-14-04056-t001:** Chemical composition of 316 L austenitic stainless steel powder.

Chemical Composition, wt%								
Cr	Ni	Mo	Mn	Si	S	P	C	Fr
17.20	12.32	2.02	0.43	0.89	0.03	0.028	0.027	balance

**Table 2 materials-14-04056-t002:** The conditions used during tribological tests.

Wear Test Conditions	
Temperature, T	23 °C
ball	Al_2_O_3_
ball diameter, d	3.175 mm
load applied, *F_n_*	5 N
friction track diameter, r	5.0 mm
sliding speed, *v*	0.1 m/s
total sliding distance, *L*	1000 m
test duration, *t*	10,000 s

**Table 3 materials-14-04056-t003:** The results of compression and tensile tests for sintered materials.

Sintered Materials	Tensile Strength R_m_ [MPa]	Elongation A_10_ [%]	Compressive Strength σ_c_ [MPa]		
			Room Temperature	400 °C	800 °C
316 L steel	690	32	774	545	167
steel + 5%ZrB_2_ (Turbula)	568	14.3	919	608	234
steel + 10%ZrB_2_ (Turbula)	530	21	980	639	245
steel + 5%ZrB_2_ (planetary mill)	832	21.7	1164	837	317
steel + 10%ZrB_2_ (planetary mill)	730	15.7	1238	848	326

**Table 4 materials-14-04056-t004:** The results of abrasive wear resistance tests.

Sintered Materials	Coefficient of Friction µ [-]	Specific Wear Rate W_v_ [mm^3^/Nm]
316 L steel	0.64	32.5 × 10^−6^
steel + 5%ZrB_2_ (Turbula)	0.55	24.6 × 10^−6^
steel + 10%ZrB_2_ (Turbula)	0.47	16.9 × 10^−6^
steel + 5%ZrB_2_ (planetary mill)	0.51	19.4 × 10^−6^
steel + 10%ZrB_2_ (planetary mill)	0.40	13.1 × 10^−6^

## Data Availability

The data underlying this article will be shared on reasonable request from the corresponding author.

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
