# Peer review of "Fabrication of the Zirconium Diboride-Reinforced Composites by a Combination of Planetary Ball Milling, Turbula Mixing and Spark Plasma Sintering"

_materials, 2021, doi:10.3390/ma14144056_

Round 1
Reviewer 1 Report
The manuscript deals with the study of the ZrB2-stainless steel composite prepared by two different milling tools: Turbula and planetary mill. The main result regarding the optimization of the mechanical properties obtained upon ball milling the powders upon planetary ball milling. The powders has been then sinterized and densified by SPS. Microstructural characterization also reveals the effect of high energy ball milling on the pre-sintered powders.
The article is well organized and does not result hard to read. The results quite interesting. However, the authors should clarify the point concerning the effect of ball milling: in the manuscript they report the difference between the milling by using two different milling tools. No data are instead available/showed varying milling energy and/or milling time, which can be considered commonly used parameters for corroborating the effect of mechanical processing. For this reason, my suggestion is to report data about the powders milled, for example, at different milling times. otherwise, my suggestion is to change the title and/or the "message" of the whole manuscript.
Please, also pay attention to the the sentence reported in the fourth page "In the study of the Young.......was used" which is reported with a different format respect to the other part of the text.
The manuscript can be accepted after the correction of these amendments.
Author Response
The manuscript deals with the study of the ZrB2-stainless steel composite prepared by two different milling tools: Turbula and planetary mill. The main result regarding the optimization of the mechanical properties obtained upon ball milling the powders upon planetary ball milling. The powders has been then sinterized and densified by SPS. Microstructural characterization also reveals the effect of high energy ball milling on the pre-sintered powders.
The article is well organized and does not result hard to read. The results quite interesting. However, the authors should clarify the point concerning the effect of ball milling: in the manuscript they report the difference between the milling by using two different milling tools.
Authors were studied the effect of method of powder mixture preparation (mixing in Turbula or milling in a planetary mill) on the microstructure and resultant physical and mechanical properties of the steel+ZrB2 composites.
No data are instead available/showed varying milling energy and/or milling time, which can be considered commonly used parameters for corroborating the effect of mechanical processing. For this reason, my suggestion is to report data about the powders milled, for example, at different milling times. otherwise, my suggestion is to change the title and/or the "message" of the whole manuscript.
The powder mixtures were produced in a ball milling only one milling condition:
- the rotational speed of the grinder - 200 rpm
- the milling time - 8 hours.
The results obtained so far have indicated that the combination of milling in a planetary mill and SPS method allows to improve the properties of composite materials.
Currently, the authors are investigating the effect of variable milling energy and milling time on the properties of steel-ZrB2 composites and they will be the subject of the next article.
The title of manuscript is corrected.
Please, also pay attention to the the sentence reported in the fourth page "In the study of the Young.......was used" which is reported with a different format respect to the other part of the text.
It is corrected.
The manuscript can be accepted after the correction of these amendments.
Reviewer 2 Report
From an industrial point of view, composite materials with ZrB2 are an important material, and authors manufactured the sintered bodies by using 2 different mixing processes. Many experimental results are presented in this manuscript. However, it is obvious that milling is better than mixing as pretreatment. While I appreciate the effort of the work presented, I think the authors needs to improve the focus of the paper. The manuscript looks like too much a summary of technical reports.
Apart from that, the following items should be revised.
/ The notation method for equipment should be described in the same way, such as SEM, Panametoric detector, Turbula.
/ * does not seem to be used in physical units.
/ Author should define “the grain size”. Size of ZrB2 or size of 316L, and show how to measure the size? Especially for lamellar shaped grain.
/ There is no “Discussion” in the manuscript.
Author Response
From an industrial point of view, composite materials with ZrB2 are an important material, and authors manufactured the sintered bodies by using 2 different mixing processes. Many experimental results are presented in this manuscript. However, it is obvious that milling is better than mixing as pretreatment. While I appreciate the effort of the work presented, I think the authors needs to improve the focus of the paper. The manuscript looks like too much a summary of technical reports.
Thank you very much for your attention. The information available in the literature on the SPS sintering of steels- ZrB2 is very scarce, especially as regards detailed analysis of the microstructure and mechanical and tribological properties of these materials. This is the first study by the authors to demonstrate the impact of the powder mixture preparation method on properties of sintered composites. At this stage of the research, one variant of the milling condition was used in a planetary mill (rotational speed of the grinder -200 rpm and the milling time - 8 hours). The results showed that the use of the milling method allowed to produce composite materials with better properties. Therefore, in the next stage of the research, the authors wiil be took into account the influence of varying milling energy and milling time also the characteristics of powders after the milling process. These results will be the subject of a separate article.
Apart from that, the following items should be revised.
/ The notation method for equipment should be described in the same way, such as SEM, Panametoric detector, Turbula.
It was corrected.
/ * does not seem to be used in physical units.
Thank you very much for your attention. The symbol * was deleted in the manuscript.
/ Author should define “the grain size”. Size of ZrB2 or size of 316L, and show how to measure the size? Especially for lamellar shaped grain.
Thank you very much for your attention. The authors made a nomenclature error. The particle size (for powders) were named mistakenly as the grain size. It is was corrected in the manuscript. The particle size of the powders is the information given on the quality certificates for the purchased commercial ZrB2 and 316L steel powders. After the milling process in the ball mill, the particle size of the composite powders was not determined.
The size of ZrB2 reinforcing phase after sintering process were determined using specialistic software. It consisted in calculating the area of each grain in pixels, and then converting the surface of the grain to the area of the circle.
/ There is no “Discussion” in the manuscript.
Discussion of results is carried in part of the article: Results and discussion.
Reviewer 3 Report
- The English of the paper needs to be improved, as there are many grammatical mistakes.
- The introduction nicely describes the available literature on the metal matrix composite. However, the ZrB2 reinforcing material’s role is not fully reviewed. There should be at least one paragraph describing the process.
- Figure 3 d and e is missing.
- The method of grain size measurement needs to be mentioned in the paper.
- In Figures 6-9 two different morphologies of borides can be observed. Also, in Figure 7 you can observe that the Mo intensity is significantly lower than the conditions in Figures 6,8 and 9. Why is that? It is mentioned that a liquid phase is formed during the milling process. Some observations needs to elaborate that.
- There needs to be explanations defining the morphologies and character of the precipitates using the EBSD analysis. Figure 11 b and c are the same (just superimposed images). Same goes for Figure 10 b and c.
- Besides, when describing the microstructure the corresponding figure needs to be mentioned in the text.
- One aspect of mechanical properties is the ductility of the developed material. It is observed in Figure 15 that the tensile data is provided. However, it would be better to demonstrate it using true stress true strain curves.
- In compression results, can you provide images of the compression samples images before and after the tests? Also, why compression tests at 400 and 800 °C?
- It is better to provide a summary of the compression and tensile results in a separate table.
- Comparing Figure 15 and 14 a significant anisotropy can be observed for the turbula and planetary mill condition (or at least it looks like it, better to show the data in a separate table). Why is that?
- The obtained results on the wear tests is better to be presented in a separate table.
Author Response
- The English of the paper needs to be improved, as there are many grammatical mistakes.
The article was again reviewed by a native speaker.
- The introduction nicely describes the available literature on the metal matrix composite. However, the ZrB2 reinforcing material’s role is not fully reviewed. There should be at least one paragraph describing the process.
Changes were made
- Figure 3 d and e is missing.
Figures are added to the manuscript.
- The method of grain size measurement needs to be mentioned in the paper.
The particle size of the powders is the information given on the quality certificates for the purchased commercial ZrB2 and 316L steel powders. After the milling process in the ball mill, the particle size of the composite powders was not determined. The size of ZrB2 reinforcing phase after sintering process were determined using specialistic software. It consisted in calculating the area of each grain in pixels, and then converting the surface of the grain to the area of the circle.
In Figures 6-9 two different morphologies of borides can be observed. Also, in Figure 7 you can observe that the Mo intensity is significantly lower than the conditions in Figures 6,8 and 9. Why is that? It is mentioned that a liquid phase is formed during the milling process. Some observations needs to elaborate that.
Authors are not mentioned in manuscript that a liquid phase is formed during the milling process.
Different morphology of borides is related to the powder mixture preparation method. The borides were fragmentation during milling in a ball mill. In addition, the use of the SPS process influence on the microstructure of the sintered composites. In the steel-ZrB2 composites were observed formation of phases containing the chromium, nickel or molybdenum. The applied conditions of SPS favor the diffusion of boron from the ZrB2 phase to the steel matrix. The austenitic matrix is a solid solution of Cr, Ni, Mo in iron. The phenomena that occur during the SPS process on the boundary of powder particles (electrical discharges, local increase in temperature, surface cleaning, vaporization, plastic flow) facilitate boron diffusion from the ZrB2 phase into the matrix, where complex borides appear.
- There needs to be explanations defining the morphologies and character of the precipitates using the EBSD analysis. Figure 11 b and c are the same (just superimposed images). Same goes for Figure 10 b and c.
I thank you very much for your attention. Figures 10c and 11c are deleted.
The analyses of orientations maps was performed with the TSL OIM Analysis software using a grain tolerance angle of 5 degrees. From orientation maps were calculated parameters like grain size, misorientation orientation and texture. The phases were next identified by the EBSD analysis (TSL OIM software and a PDF ICCD 2011 database format). The phase analysis by EBSD has indicated the presence of very fine precipitates containing chromium and boron or molybdenum or nickel , that were formed in the composites sintered by the SPS. This may suggest the formation of complex borides during the SPS sintering process carried out under some specific conditions. In this paper I did not carry out precise identification of complex borieds or another phases. A detailed analysis phase identification (TEM with with EDX and STEM) will be presented in the next article.
- Besides, when describing the microstructure the corresponding figure needs to be mentioned in the text.
It was corrected
- One aspect of mechanical properties is the ductility of the developed material. It is observed in Figure 15 that the tensile data is provided. However, it would be better to demonstrate it using true stress true strain curves.
The tensile data were summarized in Table 3.
- In compression results, can you provide images of the compression samples images before and after the tests? Also, why compression tests at 400 and 800 °C?
The images of the compression samples were added in the manuscript (Figure 14).
The determination of mechanical properties is a very important aspect of the studies of composite materials. Literature offers little information about the micro-scale, high-temperature testing of the mechanical properties of composite materials. Modern sintering techniques often produce very small samples. Sintered samples (SPS) were of 20 mm diameter and 6 mm height. The compression tests were carried at evaluated temperatures (400 and 800 °C), since the target application of the sintered composites may be operation at high temperatures. Currently, detailed tests of mechanical properties are carried out at evaluated temperatures (also using tensile tests).
- It is better to provide a summary of the compression and tensile results in a separate table.
Table 3 was added.
- Comparing Figure 15 and 14 a significant anisotropy can be observed for the turbula and planetary mill condition (or at least it looks like it, better to show the data in a separate table). Why is that?
The data are presented in a separate table.
- The obtained results on the wear tests is better to be presented in a separate table.
Table 4 was added with results of the wear tests.
Round 2
Reviewer 2 Report
Photo for Figure1 b) is missing.
Author Response
Figure 1b is added.
Reviewer 3 Report
All the requested changes are made. The paper is good to go.
Author Response
The article was again reviewed by a native speaker. Minor grammatical corrections were made.